# Effects of High-Intensity Ultrasound Pretreatment on Structure, Properties, and Enzymolysis of Soy Protein Isolate

**DOI:** 10.3390/molecules24203637

**Published:** 2019-10-09

**Authors:** Fei Zhao, Xuemei Liu, Xiuzhen Ding, Haizhou Dong, Wentao Wang

**Affiliations:** 1Department of Food Science and Engineering, Shandong Agricultural University, Tai’an 271018, China; feizhaozhaofei@126.com (F.Z.); xzd@sdau.edu.cn (X.D.); 2Engineering and Technology Center for Grain Processing of Shandong Province, Tai’an 271018, China; 3Jinan Fruit Research Institute, All-China Federation of Supply and Marketing Co-operatives, Jinan 250014, China; liuxm0218@163.com; 4State Key Laboratory of Biobased Material and Green Papermaking, Qilu University of Technology, Shandong Academy of Sciences, Jinan 250353, China

**Keywords:** high-intensity ultrasonication, soybean protein isolates, near-infrared spectra, ζ-potential, dynamic and static light scattering, antioxidant activity

## Abstract

The objective of this study was to investigate the effects of different high-intensity ultrasonication (HIU) pretreatment on the structure and properties of soybean protein isolate (SPI) as well as enzymatic hydrolysis of SPI by bromelain and antioxidant activity of hydrolysates. The HIU-treated SPI fractions showed a decrease in the proportion of α-helices and β-turns and an increase in the content of β-sheets and random coils based on Fourier-transform infrared spectroscopy. Near-infrared spectra and fluorescence spectra analyses provided support for the changes in secondary and tertiary structures of SPI after ultrasound treatment. The particle size of SPI decreased from 217.20 nm to 141.23 nm and the absolute zeta potential increased. Scanning electron microscopy showed that HIU treatment changed apparent morphology. Dynamic and static light scattering of ultrasonicated samples showed that SPI structure had changed from hard-sphere to hollow-sphere or polydisperse and monodisperse gaussian coils. HIU pretreatment significantly increased the hydroxyl-radical scavenging and the degree of hydrolysis of the SPI hydrolysates.

## 1. Introduction

Protein is an important component of human nutrition and proteins play different roles in functional food. Soybean protein isolate (SPI) has been extensively used in functional and nutritional food ingredients on account of its low price and high nutritional value. However, SPI has compact tertiary and quaternary structures, which leads to poor functional characteristics of the protein [1]. The functional properties are “the chemical and physical properties that affect the interaction between protein and other compounds in the processing, storage, and consumption” [2,3]. These properties are influenced by various factors: heating, pH, ionic strength, and chemical, physical, or enzymatic modification. Enzymatic modification is a safe and acceptable method to improve physiological activity of SPI. Single-modification treatment does not have significant efficiency. Most studies focus on combining various modification methods to improve the functional features of proteins [4,5,6]. 

Ultrasonic application of biopolymer modification has been studied thoroughly. Ultrasound technology, a novel non-thermal technology, is simple, energy saving, cost-effective, and environmentally friendly [7]. Recently, the application of ultrasound technology to improve the efficiency of enzymolysis of protein peptides has been a major focus of research [5]. The circular mechanical motions of a probe can transmit ultrasonic energy to fluid media and cause small, growing bubbles to form. Ultrasound technology has been used to mitigate the weakness of traditional enzymolysis [4]. Acoustic cavitation, which results from the interaction between bubbles and sound waves in liquids, is considered an essential effect that is mainly caused by the initiation of sonochemical reactions in liquids [8]. The effect of ultrasound is connected with heating, cavitation, turbulence, shear stresses, and dynamic agitation [9]. Ultrasound can produce highly reactive free radicals from water molecules (H_2_O → H + OH) resulting in reactions with other molecules [10]. Many studies have found that high-intensity ultrasonication (HIU) pretreatment is beneficial for increasing the rate of hydrolysis [5,11]. Some researchers found that HIU pretreatment could change the spatial structure of substrate proteins into stretched and loosened conditions, exposing the interior active sites of proteins [5,12]. Many researchers have studied how ultrasound affects the functional properties of animal and vegetable proteins and determined clear effects of ultrasound treatment on solubility, foaming, emulsifying, structure, and other functional properties of these proteins [5,6,11,13]. Wang et al. reported that the content of highly active antioxidant peptides increased in ultrasound-treated soybean β-conglycinin and glycinin because ultrasonication exposed certain groups of SPI [11]. However, little is known about the effects of high intensity ultrasound (20 kHz, 600–2000 W, 25 mm diameter titanium probe) power and time pretreatment on the structure of SPI fractions or on the antioxidant activity of their hydrolysates. 

The objective of this study was to investigate the effects of combined HIU pretreatment and enzymatic hydrolysis using bromelain on the degree of hydrolysis and antioxidant activities of SPI hydrolysates. Moreover, the effect of ultrasound power and time on structural conformations of SPI was investigated using near-infrared (NIR) spectra, Fourier-transformed infrared (FTIR) spectra, fluorescence spectra, particle size, ζ-potential, scanning electron microscopy and dynamic and static light scattering. Changes in secondary and tertiary structures, particle size, zeta potential, and enzymolysis of SPI were investigated. This work will help the study on the relationship between the structure and properties of SPI induced by HIU treatment. Dynamic and static light scattering analysis had revealed the changes in aggregation behavior of SPI induced by ultrasound, which led to the unfolding of the protein structure and formation of soluble aggregates. We studied the enhancement of enzymatic hydrolysis and antioxidant activities of SPI hydrolysates due to HIU treatment. The results of this work will be useful for the development of ultrasound technology in the protein industry. 

## 2. Results and Discussion

### 2.1. Near-Infrared Spectroscopy Analysis

Near-infrared spectroscopy (NIR) is a noninvasive, nondestructive, fast, and precise analytical method that is widely used in the pharmaceutical and food industries [14,15,16]. Nondestructive NIR can be used to study protein secondary structure and generate spectral data of samples in different physical conditions [17]. Figure 1A described the changes in NIR spectra of SPI according to characteristic shifts in some band frequencies [18]. SPI showed a specific peak (the double frequency band) at 9814 cm^−1^ (N−H stretching vibration of the protein group). The primary frequency bands of SPI are at 6653 (N−H stretching vibration), 5876 (C−H stretching vibration), and 5765 cm^−1^ (C−H stretching vibration). SPI also exhibited large water absorption bands at 5148 cm^−1^ (O−H stretch and H−OH deformation combination). The absorption band of combined S−H stretching and bending is at 5071 cm^−1^; the combination of amide A and amide II is at 4863 cm^−1^; the combination of amide I and amide III is at 4597 cm^−1^; the C−H stretch and deformation combination is at 4348, 4261, and 4044 cm^−1^. HIU-treated SPI slightly reduced intensities of the characteristic peaks, such as 6653, 5148, 5071 cm^−1^, and the intensities of partial absorption peaks (5876, 5765, 4863, 4596, 4347, and 4261 cm^−1^) were increased. HIU-treated SPI mostly lacked the water-related band (5148 cm^−1^) and partial HIU-treated sample reduced intensities of these water-related bands, which were shifted to lower frequencies by 4 and 13 cm^−1^, with respect to the control. HIU treatment led to significant red shifts of the SH-related bands (about 7−39 cm^−1^) and the peak intensities of SH-related bands decreased upon the increase of ultrasound power and time. The result indicated that the content of the sulfhydryl group in SPI reduced after HIU treatment and formed disulfide bond [19]. The band of amide I and amide III combination was shifted to lower wave-number (about 2−14 cm^−1^) after HIU treatment. This was due to the formation of hydrogen bands. HIU treatment also led to red shifts of C−H stretching vibration (about 1–4 cm^−1^) and C−H stretch and deformation combination (about 1–7 cm^−1^). HIU treatment led to blue or red shifts of N−H stretching vibration; the absorption peaks of 600 W, 800 W, 1000 W, and 1200 W HIU-treated samples were shifted to lower wave-numbers (6645–6651 cm^−1^), and those of 1400 W, 1600 W, 1800 W, and 2000 W HIU-treated samples were shifted to higher wave-numbers (6654–6666 cm^−1^). 

Particular secondary structures of protein NIR spectroscopy absorption bands can be obtained by second-derivative or deconvolution techniques. The NIR spectra of proteins showed several bands at similar frequencies in the combination (4000–5000 cm^−1^) and first overtone (5600–6600 cm^−1^) spectral regions. Some bands suggested α-helix (4099, 4372, 4614, and 5760 cm^−1^) and β-sheet (4068, 4322, 4406, 4528, 4864, and 5917 cm^−1^) structures based on deconvolution NIR spectra of SPI. These band positions were consistent with previous studies [14,20]. Figure 1B–D showed the vector-normalized second-derivative near-infrared spectra of proteins (4000–5000 cm^−1^) and the changes in NIR spectra of SPI according to characteristic shifts in the frequencies of some bands. The absorption bands of SPI with HIU treatment were obviously different from those of untreated SPI. HIU treatment led to a significant red shift of the β-sheet structure band peaks of SPI (4068 cm^−1^ and 4528 cm^−1^ about 1 cm^−1^; 4322 cm^−1^ about 1–3 cm^−1^; and 4864 cm^−1^ about 2–3 cm^−1^), and blue shifts of 4406 cm^−1^ (about 1–3 cm^−1^). HIU treatment also led to a significant blue shift of the α-helix structure band peaks of SPI (4099 cm^−1^ about 1–3 cm^−1^; 4372 cm^−1^ about 2–3 cm^−1^; 4614 cm^−1^ about 1–2 cm^−1^). This result showed that the change of secondary structure can be ascribed to the fact that protein molecular interactions were destroyed by HIU treatment. The interactions may include interactions between diverse parts of the protein molecule, such as disulfide bonds and between local amino acids sequence [19].

### 2.2. Fourier-Transform Infrared Spectroscopy Analysis

FTIR spectra of SPI were recorded to illuminate the secondary structure changes of SPI after HIU pretreatment. Figure 2A shows the FTIR spectra of SPI. The absorption bands of SPI with HIU treatment were different from those of untreated SPI. The main characteristic spectrum consisted of three intense bands from amide I (1700–1600 cm^−1^), amide II (1560–1520 cm^−1^), and amide III (1430–1240 cm^−1^) [21]. Compared to untreated-SPI, the intensity increase of the absorption peaks of HIU-treated SPI were more obvious between 4000 and 400 cm^−1^. The band of amide I was shifted to a lower wave-number (about 2 cm^−1^) after HIU treatment. The result was in accordance with that of the NIR spectra. This was due to the formation of hydrogen bands and the bonding constant of the N−H bond decreased, then the vibration dipole moment of the corresponding group increased. The amide II (about 4 cm^−1^) and III (about 1 cm^−1^) band peaks of SPI generated blue shift after HIU treatment. A similar observation was got by Zhou et al. for corn gluten meal [22]. This result indicated that the secondary structure of SPI was changed after ultrasound pretreatment, which might be because the interactions in protein molecules were broken down by ultrasonication. 

The amide I (1700–1600 cm^−1^) band was used to identify and calculate the secondary structures of SPI [21]. Characteristic absorption peaks of SPI were determined by deconvolution and Gaussian curve; these include β-turn (1700–1660 cm^−1^), α-helix (1660–1650 cm^−1^), random coil (1643 cm^−1^), and β-sheet (1640–1610 cm^−1^) [3,21]. The peak areas were calculated to identify the percentage of the secondary structures. The effects of ultrasonic power and time on the composition of secondary structures were listed in Figure 3. Compared with untreated-SPI, the β-sheet, and random coil contents of HIU-treated SPI increased, while the α-helix and β-turn content decreased. This observation indicated that the α-helix and β-turn were transformed to β-sheet and random coil. Other researchers reported similar results [13,23]. Generally speaking, the amide I band of an IR spectrum is ascribed to C=O stretching vibrations with certain C−H stretching and N−H bending patterns. The C=O stretching vibrations mostly rely on diverse secondary structures and inter- or intramolecular effects (molecular structure model and hydrogen bonding mode), which makes the amide I band the most sensitive IR spectral region to indicate secondary structural information of proteins [24].

Divergent results in changes of ultrasonic SPI secondary structure content were also observed in our study. For instance, α-helix and random coil contents of SPI were reduced and their β-turn content increased by HIU treated at 800 W (5 min), 1600 W (5 min), 1800 W (30 min), and 2000 W (20 min). A similar observation was reported by Huang et al. for SPI [25]. However, some researchers reported that cavitation shearing might disrupt the tertiary structure of soybean β-conglycinin and glycinin but leave most secondary structural elements intact [13]. Zhou et al. found that HIU influenced glycinin aggregates but secondary and tertiary structures were retained [22]. These disparate results might be attributed to the differences in sonication conditions and native proteins. The shear forces of ultrasound mechanical action led to differences in secondary structure and broke down the interactions in protein molecules and affected the protein molecule internal structure [26].

The changes of the microenvironment in aliphatic amino acid residues was measured by the variation in spectra at 1310 cm^−1^ as well as at 1449 cm^−1^, at 2931 cm^−1^, at 2960 cm^−1^ (Figure 2B,C) [21]. The C−H_3_ stretching vibration (2960 cm^−1^) band intensity had no obvious change, while 1310 cm^−1^ (C−H_2_ twisting vibration) and 1449 cm^−1^ (C−H_3_ asymmetric variable angle vibration) band spectra had blue shift (about 1 cm^−1^), 2931 cm^−1^ (C−H_2_ stretching vibration) band spectra had red shift (about 2 cm^−1^). HIU treatment increased spectral absorption at 3074 cm^−1^ (amide B, about 4 cm^−1^), 1154 cm^−1^ (C=S stretching, about 2 cm^−1^), and at 1078 cm^−1^ (C−C stretching, about 2 cm^−1^) (Figure 2B,C). HIU treatment also led to a significant red-shift (about 4–8 cm^−1^) of amide A (3298 cm^−1^, N−H bending or O−H stretching vibration) band peak of SPI (Figure 2B), showing that more N−H groups participated in hydrogen bonding in polypeptide chains [27]. Shifting of FTIR spectrum peaks was an indirect indication of HIU-induced unfolding of protein structures. HIU treatment might modify the structure of the protein by exposing the auxochrome and chromophore groups. The auxochrome group includes −SH, −OH, and −NH_2_, and the chromophore group includes N=N, N=O, C=O, C=C, and −COOR. Protein secondary structure could unfold and detach during the ultrasonic treatment process. Theoretically, sample pretreatment, such as HIU treatment could make the protein denaturation and unfold the active sites of the protein. This could explain the particle size change of SPI induced by ultrasonic treatment leading to exposure of hydrophobic and polar groups.

### 2.3. Fluorescence Spectra Analysis

To investigate the effect of HIU on the conformation of SPI, intrinsic fluorescence spectra of untreated and HIU-treated SPI were obtained. When conformational changes of SPI occur, the fluorescence spectra of a protein changes according to the local molecular environment of the tryptophan (Trp), tyrosine, and phenylalanine groups [28]. Significant changes in the tertiary structure of SPI induced by HIU pretreatment were apparent from changes in maximum wavelength (λ_max_) and fluorescence intensity (Figure 4). Generally, if the λ_max_ is > 330 nm, then Trp is designated as a “polar” environment; if λ_max_ < 330 nm, then Trp is designated as a “non-polar” environment [29]. In this paper, Trp of all samples were determined to be present in a “polar” environment. The λ_max_ of HIU-treated SPI had shorter wavelengths (hypochromic shift) (from 343 nm to 336–339 nm), which indicated a change in the polarity of the microenvironment of Trp residues. This might be due to the fact that HIU destroyed the internal hydrophobic interactions of protein molecules, exposing the chromophore and auxochrome groups. The intrinsic fluorescence intensity of ultrasonication-treated SPI distinctly increased compared with that of the control and higher sonication time caused more change in fluorescence intensity. Similar observations were reported for dephenolized sunflower meal after ultrasound treatment [30]. The increase in fluorescence intensity by HIU treatment might be attributed to the exposure of the hydrophobic interactions of protein molecules and exposure of the internal Trp residues. Pallares et al. reported that unfolding of proteins resulted in exposure of more chromophores to the solvent, which altered fluorescence intensity [31]. The results indicated that HIU pretreatment affected the force of hydrophobic interaction and the tertiary structure of SPI.

### 2.4. Effective Diameter Analysis

The particle size is one of the factors that influences the functionalities of protein [13]. In this study, HIU treatment significantly reduced the particle size of SPI (Figure 5). The effective diameter of the SPI dispersions varied between 141.23 nm and 193.33 nm, and in all cases were significantly lower than that of the control (217.20 nm). Reduction in particle size after ultrasound treatment was also reported for pea protein isolate, sunflower protein isolates, and fava bean protein [30,32,33].

Low-power (<1000 W) ultrasound treatment for a short time (<15 min) led to smaller SPI particles. Higher ultrasonic power with relatively long treatment times significantly increased SPI particle size, implying the formation of small aggregates. The mean particle size of SPI was 181.31 nm after HIU treatment at the ultrasonic power of 2000 W. Zhou et al. reported that longer HIU time (40 min) led to reaggregation of glycinin, HIU induced simultaneous dissociation, and aggregation of glycinin aggregates [23]. Martínez-Velasco et al. found that the smaller protein particles were produced at high amplitudes and short times, while the larger protein particles were formed by applying low amplitudes for relatively long times [33]. The effective diameter changes induced by long-duration ultrasonic treatment was referred to as “over-processing” [34]. Following low-power ultrasonic treatment, protein particles were violently broken up into smaller soluble protein aggregates by the ultrasonic cavitations. When the power of ultrasonic treatment was increased, the shear, turbulence, and micro-streaming forces of ultrasound mechanical action changed protein secondary structure and increased collision and aggregation speeds. Protein aggregates might form due to noncovalent interactions, such as hydrophobic interactions [35]. The changes in hydrodynamic diameters of proteins could be due to hydrodynamic shear forces related to ultrasonic cavitations and disruption of associative electrostatic and hydrophobic interactions. 

### 2.5. Effects of HIU Treatment on ζ-Potential of SPI

ζ-potential is an index of the surface charge characteristic of protein particle in solution. Generally, the positive net electronic charge is mainly due to lysine and histidine acid and the negative net charge attributes to aspartic and glutamic acid [36]. If more negatively charged amino acids are present than positively charged amino acids, the ζ-potential of a protein solution is negative [37]. From Figure 6, the ζ-potentials of the samples after low-power and short time (<15 min) had more positively charged amino acids than negatively charged amino acids. With the increase of ultrasonic power and time, the ζ-potentials of samples were negative. Our results showed that the highest ζ-potential was got at 1600 W (20 min) treated sample, due to the increase in exposure of anionic groups on the SPI surface, and the ζ-potential followed by a decrease when the ultrasonic power and time increased. The effective surface charge of the SPI particles mainly decides their aggregation and dispersion [38]. Ultrasonication at medium and high power might increase the negative surface charge on proteins, improve the electrostatic repulsions of SPI interparticle, damage existing SPI aggregates, control further aggregation, and increase the stability of the protein dispersions [29]. This phenomenon could improve the solubility of SPI. The decrease in the absolute zeta potentials of samples was due to the formation of SPI aggregates when the samples suffered from a high-powered and prolonged sonication.

### 2.6. Scanning Electron Microscopy

The effects of different HIU treatments on the microstructure of SPI were investigated by scanning electron microscopy (SEM). Figure 7 showed a set of SEM images of SPI samples with 200-fold magnification. After HIU treatment with various power and time, the spherical structure of SPI with the wrinkled surface (B: 600 W 5 min, C: 1000 W 25 min, D: 1800 W 5 min) disappeared, and the surface became fine and smooth, exhibiting more irregular and smaller fragments. Changes in protein structure were due to the effect of micro-streaming and turbulent flow produced by during the ultrasonic processing. SPI gathered randomly to form layer block structure in the vacuum freeze-dry process, effectively breaking the dense structure of the protein and becoming friable in texture. The structure of the black bean protein and the pea isolated protein were also changed by ultrasonic treatment [29,32]. The fragment of sample D was smaller and more homogeneous than that of sample B and C, showing that the high-power HIU treatment had a stronger effect on the protein. When the power of the ultrasonic treatment was increased, the particles became smaller [10,39]. The results may be attributed to the changes in unfolding of the protein molecules caused by ultrasonic treatment and improved exposure of hydrophobic groups (Section 3.3) to the molecular surface and formation disulfide bonds (Section 3.1) between the molecules, which could cause interaction in the protein molecules and form layer block aggregates during freeze-drying process. 

### 2.7. Light Scattering Analysis

The light scattering technique includes static and dynamic light scattering, which can be determined from the structure and size of large aggregates. The molecular parameters in solution got from static light scattering (SLS), such as weight-average molecular weight (M_w_), the z-average mean radius of gyration (R_g_), and the second virial coefficient (A_2_). The angular dependence of hydrodynamic radius (R_h_) obtained from dynamic light scattering (DLS). Light scattering is a rapid and efficient method for the determination of the protein aggregation behavior [40]. The light scattering data of SPI samples were shown in Table 1.

From Table 1, there was a significant increase in the M_w_ of all SPI samples with different HIU treatment. Globular protein changed into non-spherical particles and M_w_ of protein particles in the solution increased during ultrasonic treatment, which presented a “large particle” phenomenon. This was due to the formation of hydrogen bonds and hydrophobic interactions of protein molecules caused by ultrasonic treatment. Tang et al. reported that the changes in partical size of the protein after ultrasound treatment might be attributed to the formation of soluble aggregates [41]. The M_w_ of SPI (600 W) was larger than other samples. Table 1 showed that HIU treatment at 600 W (30 min) remarkably increased the M_w_ of SPI from 2.89 × 10^6^ (control) to 3.11 × 10^8^ g/mol. Following low-power HIU treatment, the effect of micro-streaming and turbulent forces might increase the frequency of collision and aggregation. The ζ-potentials of SPI samples examined were positive, molecular interactions were mainly controlled by electrostatic forces, this typically caused by the formation of unstable aggregates and an increase in M_w_ of SPI. The M_w_ of SPI became smaller with the ultrasonic power increasing. This was because the ζ-potentials of samples were negative (Figure 6) after high-power HIU treatment, which indicated that molecular interactions of SPI were mainly controlled by electrostatic repulsion; the SPI aggregates were broken down and changed into smaller soluble aggregates.

The A_2_ value is a characterization of the interaction between the polymer and solvent [42,43]. The A_2_ value of untreated protein was negative, indicating that protein molecules were inclined to aggregate, and the attraction between molecules was not conducive to the dispersion of protein in water, while the A_2_ value of the SPI solution after HIU treatment was positive, which improved protein solubility in water.

Compared to untreated protein, the R_g_ of protein treated by low-power and short-time (5 min) ultrasound showed little change. This indicated that the protein-peptide chain was unfolded by ultrasonic treatment, the partial polar and hydrophobic groups were exposed, and a low amount of protein structure was damaged. The R_g_ of SPI increased with an increase of ultrasonic power and time. The results might be due to noncovalent interactions, such as hydrophobic interactions, might drive aggregation [35]. In this study, the tertiary structure of SPI measurements confirmed this view. The R_g_ of SPI were consistent with the trends of protein solubility. The hydrodynamic radius (R_h_) obtained from the DLS measurement was shown in Table 1. The R_h_ of the samples treated by low-power ultrasound were lower than the control. This suggested that the protein denatured and dissociated into small protein molecules after HIU treatment. The R_g_ of the partial SPI increased with an increase of ultrasonic power and time.

As hydrodynamic and static dimensions change with the macromolecule structure. The combination of DLS and SLS can provide architecture information of the giant molecules [42]. The parameter ρ is the ratio of R_g_ to R_h_, and is an important parameter for characterizing macromolecules structure. In general, the ρ value increases with decreasing branching density, but a decrease in polydispersity counteracts the effect of branching [40]. The ρ value of the untreated protein was 0.78, which is predicted for a hard (homogeneous) sphere. Higher ρ values were obtained for the HIU-treated SPI. The ρ value was found to be 1.0 for the SPI treated with the low-power (600 W) and short-time (5 min) ultrasonic treatment, which is predicted for hollow-sphere (a highly branched globular shape molecule). Obviously higher ρ values (1.27–2.1) were obtained for samples which were treated by medium- or high-power ultrasonic treatment. The increase in ρ value may be ascribed to the effect of less branching, although increased polydispersity may cause the different structure of SPI. The ρ value is incompatible with a polydisperse gaussian coil of linear chains (1.73 at θ-solvent and 2.05 in good solvent); it is even lower than the predicted values for a monodisperse gaussian coil, which are 1.5 at θ-solvent and 1.78 in a good solvent, respectively [44,45]. This suggested that the SPI structure changed from hard-sphere to hollow-spheres, or polydisperse and monodisperse gaussian coils, after HIU treatment. During the HIU treatment process, the globular structure of SPI was unfolded, and the polypeptide chaindepolymerized, then proteins would further be associated into different conformational aggregates via hydrogen bonding, electrostatic, and hydrophobic interaction to form macromolecules. HIU treatment led to protein flexibility enhancement in a liquid solution, and more interior hydrophobic zones of the protein were exposed. The exposed parts of SPI were various with different powers HIU treatment. Therefore, the hydration patterns of the protein would have a remarkable effect on the structural dynamic properties with different ultrasonic treatments.

### 2.8. Antioxidant Activity of SPI Hydrolysates

The effects of HIU pretreatment on hydroxyl-radical scavenging capacity of SPI hydrolysates were shown in Figure 8. Compared with untreated SPI hydrolysates, ultrasonication significantly increased the hydroxyl-radical scavenging capacity of SPI hydrolysates. The antioxidant activity of ultrasound treated SPI (1400 W, 15 min) hydrolysates were the highest in all samples. This result may be ascribed to the secondary and tertiary structural changes of SPI fractions [11]. HIU pretreatment exposed the hydrophobic interaction sites of SPI, increased the contact area between enzyme and substrates, and improved the antioxidant activity of the SPI hydrolysates. With increased ultrasonic power and time, the protein hydrophobic interaction sites that had been exposed to the surface, were obscured inside the molecules [3,29]. Enzymatic hydrolysis of protein combined with HIU pretreatment might be conducive to obtaining safe and effective active peptides. Results showed that bioactive peptides in protein hydrolysis products relied on ultrasound conditions. 

### 2.9. Degree of Hydrolysis of SPI Hydrolysates

The DH of SPI hydrolysates were shown in Figure 8. Compared with untreated SPI hydrolysates, HIU pretreatment significantly increased the DH of SPI hydrolysates. This result might be attributed to the cavitation shearing of ultrasonic waves that destroyed the secondary and tertiary structure of SPI and exposed the enzymes’ sensitive sites of protein fractions [4]. This exposure increased the contact area between enzymes and substrates and improved the extent of subsequent proteolysis [11]. The DH of HIU-treated SPI (1000 W, 25 min) hydrolysates was the highest. Compared to the control, the DH of the SPI-1000 W-25 min and SPI-2000 W-20 min hydrolysates increased by 134.4% and 130.8%, respectively. The changes in the secondary and tertiary structure of SPI might lead to the different DH of SPI hydrolysates. 

We found that the HIU-treated SPI hydrolysates had a higher DH and stronger antioxidant activity than those of untreated SPI hydrolysates (Figure 8). The phenomenon indicated that the antioxidant activity was related to the DH but not in a linear relationship. The SPI-1400 W-15 min hydrolysates had the highest antioxidant activity, while the SPI-1000 W-25 min hydrolysates obtained the highest DH. This may be because some antioxidant peptides were degraded into amino acids or smaller peptides with an increased DH [46].

## 3. Materials and Methods

### 3.1. Materials

Soybean protein isolate (protein content 90.31%) was purchased from Shandong Wonderful Industrial Group Co. Ltd. (Dongying, Shandong Province, China). Bromelain was purchased from Guangxi Pangbo biological engineering Co. Ltd. (Nanning, Guangxi Province, China). 

### 3.2. High-Intensity Ultrasound Treatment of SPI

To prepare SPI (5% *w*/*v*) solutions, SPI powder was dissolved in deionized water under stirring for 2 h at room temperature and stored at 4 °C overnight. An ultrasound processor (TJS-3000 Intelligent Ultrasonic Generator V6.0, 20 kHz, 25 mm diameter titanium probe, Hangzhou Success Ultrasonic Equipment Co. Ltd., Hangzhou, Zhejiang, China) was used to process 50 mL of SPI solutions in 100 mL glass vessels that were immersed in an ice-water bath. Ice was added every 5 min. SPI solutions were treated at 600 W, 800 W, 1000 W, 1200 W, 1400 W, 1600 W, 1800 W, and 2000 W for 0, 5, 10, 15, 20, 25, and 30 min. Samples were then lyophilized and stored for further use.

### 3.3. Near-Infrared Spectroscopy

SPI samples were scanned using a Fourier-transform near-infrared spectra analyzer (Antaris II, Thermo Fisher, Waltham, MA, USA). Samples were gently pressed into the dish (30 mm), then tapped three times with a spatula to ensure even packing. NIR spectra were captured at wavelengths of 10,000–4000 cm^−1^ (64 scans) measured as absorbance at a resolution of 8 cm^−1^ [14].

### 3.4. Fourier-Transformed Infrared Spectroscopy 

A FTIR spectrometer (Nicolet is5, Thermo SCIENTIFIC, Waltham, MA, USA) was used to analyze the samples. Samples were prepared as potassium bromide pellets. Each sample was subjected to 32 scans at 4 cm^−1^ resolution from 4000 to 500 cm^−1^. The spectra of the samples were analyzed using the Omnic software (OMNIC 8.2, Thermo Nicolet Corporation, Waltham, MA, USA) [47].

### 3.5. Fluorescence Measurements

The fluorescence spectra of ultrasound-treated and untreated SPI samples were obtained using a Lumina fluorescence spectrometer (Thermo Fisher Scientific, Shanghai, China). The ultrasound-treated and untreated SPI solutions (0.2 mg/mL) were prepared in 10 mM phosphate buffer (pH 7.0). The excitation wavelength was 290 nm and the emission spectra were recorded from 300 to 500 nm at a constant slit of 5 nm for both excitation and emission [11].

### 3.6. Particle Size Determination

The particle size of SPI samples was determined following ultrasound treatment. Prior to measuring, all samples (1.5 mg/mL) were filtered using 0.45 μm Millipore filters to remove dust. The samples were used for particle size measurements as the effective diameter and the width of the size distribution was determined as a polydispersity index with a NanoBrook ZetaPlus Potential Analyzer (Brookhaven Instruments, Holtsville, NY, USA), using a refraction index of 1.334. 

### 3.7. Zeta (ζ) Potential Measurements

The ζ potentials of the proteins were determined using a Zetapotential analyzer (ZetaPlus, Brookharen Instruments, Holtsville, NY, USA). The SPI dispersions were filtered through 0.45 μm Millipore filters and injected into the apparatus to measure the ζ potential; the averages of the three measurements are reported as the ζ potential. 

### 3.8. Scanning Electron Microscopy

The morphology of the HIU-treated SPI samples was researched by Scanning Electron Microscopy (SEM). The freeze-dried protein was placed in the copper sample-holder using conductive carbon tape and coated with gold. Then, the samples were determined with a Supra 55 electron microscopes (ZEISS, Jena, Germany) at an accelerating voltage of 5 kV.

### 3.9. Light Scattering 

Dynamic and static light scattering of SPI solutions were determined by BI-200SM dynamic laser scattering system (Brookhaven Instruments, Holtsville, NY, USA). The helium-neon laser with 35 mW power and 633 nm wavelength was as the light source. The samples prepared for DLS and SLS were filtered through 0.45 μm Millipore filters into a precision cylindrical cell (quartz, diameter: 25 mm). Light scattering measurements were measured in the angular range of 30–120° for static measurements and 30–90° for dynamic measurements [40]. The refractive index increment was determined to be 0.185 mL/g for SPI in aqueous solution. The autocorrelation function was analyzed using CONTIN and NNLS software. All the light scattering measurements were conducted at 25 °C and controlled within 0.01 °C by water circulating apparatus.

### 3.10. Preparation of SPI Hydrolysates

Untreated and ultrasound-treated SPI dispersions (5% *w*/*v*) were prepared by adding protein powder to distilled water. The SPI solutions were adjusted to pH 7.0 with 1 M NaOH and then incubated in a water bath at 55 °C. Enzymatic hydrolysis reactions were started using protein solutions with added bromelain. The enzymatic reactions were stopped by boiling water for 10 min after 3 h. The enzymatic hydrolysates were centrifuged for 10 min at 10,000× *g*. The supernatant was freeze-dried and used for further analysis. 

### 3.11. Degree of Hydrolysis

The degree of hydrolysis (DH) of SPI hydrolysates was calculated according to the pH-stat method [48]. The percent DH was calculated according to the following equation: (1)$$\mathrm{DH}\;=\;\frac{B\;\times \;N}{\alpha \;\times \;\mathrm{Mp}\;\times \;h}\;\times 100\%$$
where B is the NaOH consumption (mL), N is the concentration of NaOH (1 M), α is the degree of α-amino groups dissociation, M_p_ is the protein mass (g), and for SPI, h = 7.75 mmol/g protein.

### 3.12. Determination of Hydroxyl-Radical Scavenging Capacity

The assay was performed based on the generation of HO-radicals from a Fenton reaction between ferrous ions and hydrogen peroxide. The hydroxyl-radical scavenging capacity of SPI hydrolysates was determined according to the method of Smirnoff et al. with some modifications [49]. Freeze-dried hydrolysate samples were dissolved in distilled water to obtain 5.0 mg/mL solutions. An aliquot (2 mL) of the sample solution was mixed with FeSO_4_ (2 mL, 6 mM) and H_2_O_2_ (2 mL, 6 mM) solutions. The mixture was allowed to stand at 30 °C for 10 min, then salicylic acid (2 mL, 6 mM) was added, and the mixture was incubated at 37 °C for 30 min. The absorbance was recorded at 510 nm using a spectrophotometer. The hydroxyl-radical scavenging capacity by the SPI hydrolysates was calculated as follows:(2)$$\mathrm{Hydroxyl}\text{-}\mathrm{radical}\;\mathrm{scavenging}\;\mathrm{capacity}\;(\%)\;=\;(1-\;\frac{A_{i}{\;-\;A}_{j}}{A_{0}})\;\times \;100\%$$
where A_i_ was the absorbance of FeSO_4_, H_2_O_2_, and salicylic acid with the SPI hydrolysates; A_j_ was the absorbance of FeSO_4_, H_2_O_2_, and distilled water with the SPI hydrolysates; A_0_ was the absorbance of FeSO_4_, H_2_O_2_, and salicylic acid with distilled water. 

### 3.13. Data Analysis

Data were presented as the means ± standard deviations (SD) of three replicate determinations. The analysis of Fourier-transform infrared spectroscopy and near-infrared spectra was conducted using OMNIC software (Thermo Fisher Scientific, Waltham, MA, USA) and PeakFit software version 4.0 (Systat Software Inc, San Jose, CA, USA).

## 4. Conclusions

The effects of HIU treatment performed at various powers (600–2000 W) and lengths of time (5–30 min) caused structural and property changes in SPI samples. NIR spectra revealed that the secondary structure of HIU-treated SPI was altered, especially the peak intensities of SH-related bands decreased after ultrasonication indicating that changes in the content of the sulfhydryl group in SPI can be determined by NIR spectra quickly. HIU treatment led to striking changes in the secondary and tertiary structures of SPI determined by the FTIR spectra and fluorescence spectra. The particle size decreased and the absolute zeta potential increased after ultrasonication. The changes in molecular interactions attribute to ultrasound were further confirmed by SEM, DLS, and SLS of protein. After ultrasound treatment, the SPI samples had larger aggregates and formed into different conformational aggregates via molecular interaction. The structural changes of SPI could be of benefit to the improvement of physicochemical property and protein solubility, as well as the DH and antioxidant activity of SPI hydrolysates. DH of SPI hydrolysates (SPI-1000 W-25 min) increased by 134.4% compared to the control, and the antioxidant activity of SPI hydrolysates treated at 1400 W for 15 min was higher than that of other samples. The results of this paper could satisfy the comprehensive needs of manufactured protein products and supply the theoretical basis and direction for the SPI development, but further study of many properties are required to explore the detailed mechanism. 

## Figures and Tables

**Figure 1 molecules-24-03637-f001:**
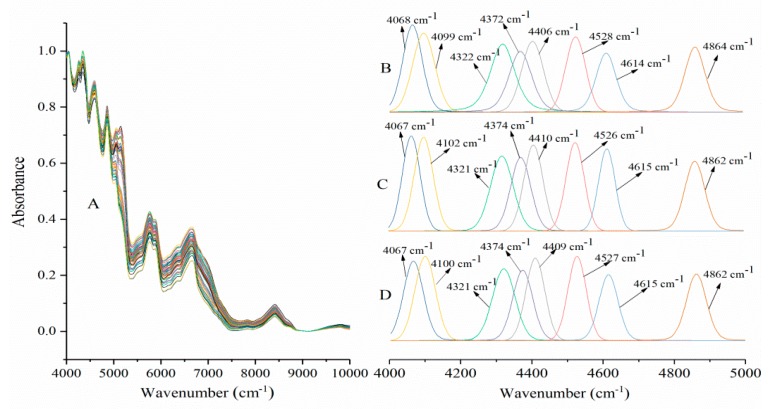
Near-infrared spectra of soybean protein isolate (SPI) after high-intensity ultrasonication (HIU) treatment from 4000 to 10,000 cm^−1^ (**A**) and fitting of peaks to SPI from 4000 to 5000 cm^−1^ (**B**: Untreated-SPI, **C**: HIU-600 W-30 min-SPI, **D**: HIU-1400 W-5 min-SPI).

**Figure 2 molecules-24-03637-f002:**
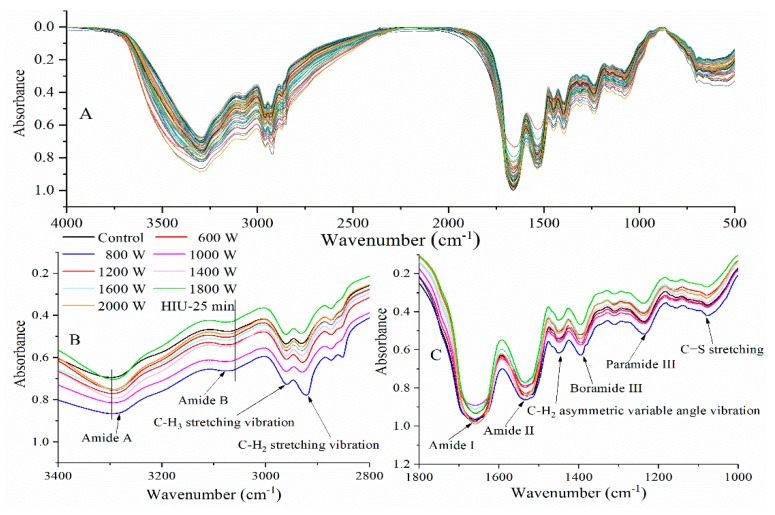
FTIR spectra of SPI after HIU treatment (**A**: 4000–500 cm^−1^) and HIU treated 25 min (**B**: 3400–2800 cm^−1^, **C**: 1800–1000 cm^−1^).

**Figure 3 molecules-24-03637-f003:**
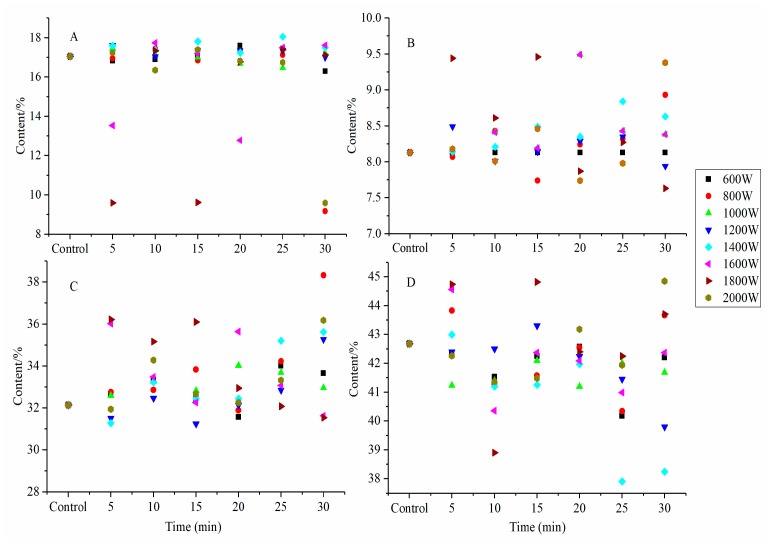
Effects of HIU treatment on secondary structure composition of SPI (**A**: α-helix, **B**: random coil, **C**: β-sheet, **D**: β-turn).

**Figure 4 molecules-24-03637-f004:**
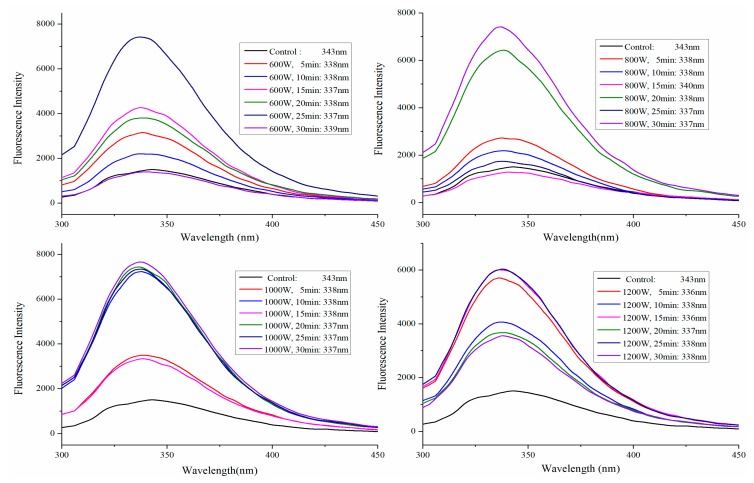
Fluorescence spectra of SPI after HIU treatment.

**Figure 5 molecules-24-03637-f005:**
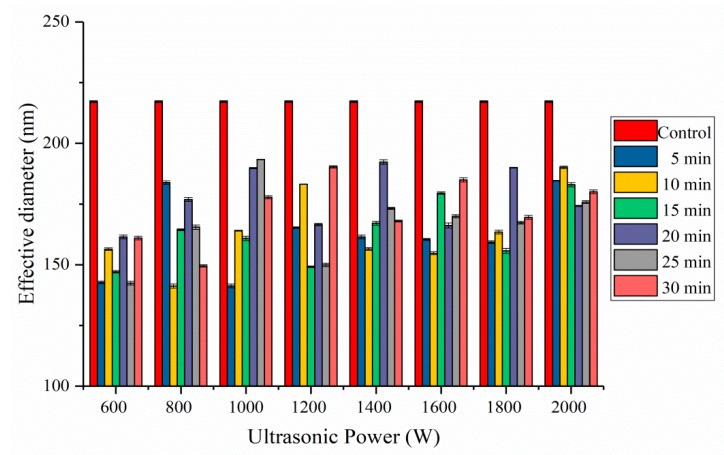
Effective diameter for SPI samples filtered at 0.45 μm. Data are the averages of three replications ± standard deviation.

**Figure 6 molecules-24-03637-f006:**
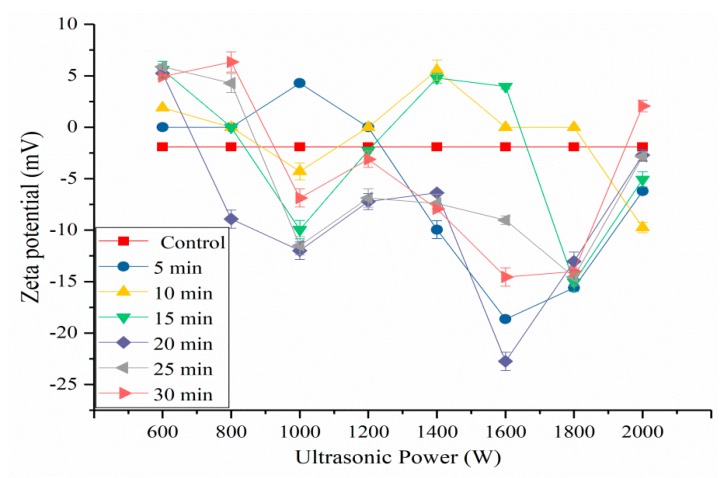
Zeta-potential value of HIU-treated SPI.

**Figure 7 molecules-24-03637-f007:**
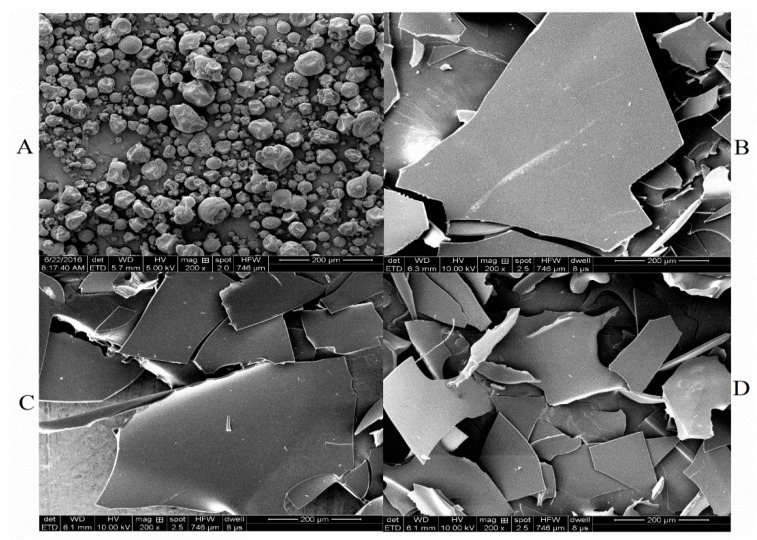
Scanning electron micrographs of SPI (**A**: non-HIU treated SPI, **B**: 600 W-5 min HIU-treated SPI, **C**: 1000 W-25 min HIU-treated SPI, **D**: 1800 W-5 min HIU-treated SPI).

**Figure 8 molecules-24-03637-f008:**
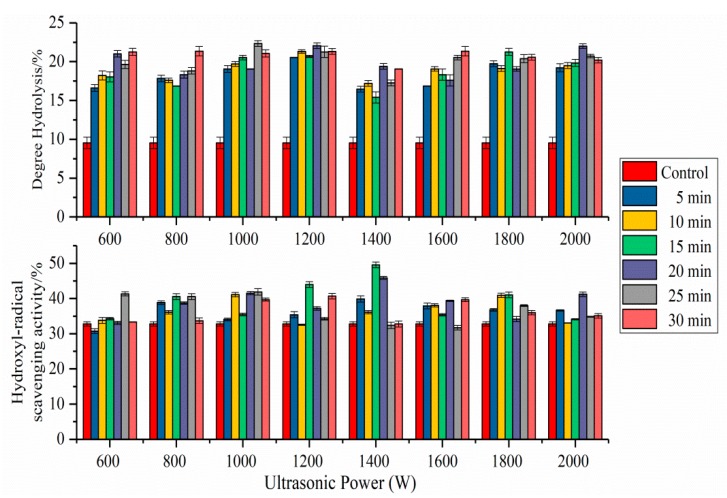
Effect of pretreatment on enzymatic hydrolysis and hydroxyl-radical scavenging activity of the SPI hydrolysates. Data are the averages of three replications ± standard deviation.

**Table 1 molecules-24-03637-t001:** Summaries of dynamic and static light scattering data on the soluble aggregates in the SPI aqueous solutions after HIU treatment.

Entry	M_w_ (g/mol)	R_g_ (nm)	R_h_ (nm)	A_2_	ρ	Structure
Control	2.89 ± 0.17 × 10^6^	96 ± 5.4	122.97	−3.28 ± 0.6 × 10^−4^	0.78	Hard-sphere
600 W-5 min	2.57 ± 0.39 × 10^8^	100 ± 18	95.99	8 ± 3.8 × 10^−7^	1.04	Hollow-sphere
600 W-15 min	8.2 ± 0.33 × 10^7^	93 ± 4.8	89.86	4.31 ± 0.8 × 10^−6^	1.03	Hollow-sphere
600 W-30 min	3.11 ± 0.2 × 10^8^	96.1 ± 7.4	90.63	4.23 ± 0.38 × 10^−6^	1.06	Hollow-sphere
800 W-5 min	5.44 ± 0.22 × 10^7^	97.5 ± 4.7	123.56	4.7 ± 1.3 × 10^−7^	0.79	Hard-sphere
800 W-15 min	1.41 ± 0.26 × 10^7^	166 ± 20	121.53	1.3 ± 0.81 × 10^−5^	1.37	GCM (θ-solvent)
800 W-30 min	3.2 ± 1.2 × 10^7^	224 ± 48	123.77	3.6 ± 1.3 × 10^−5^	1.81	GCM (good-solvent)
1000 W-5 min	1.05 ± 0.17 × 10^7^	95 ± 18	94.36	8.5 ± 1.1 × 10^−7^	1.01	Hollow-sphere
1000 W-15 min	1.90 ± 0.32 × 10^7^	157 ± 20	123.76	9.46 ± 0.8 × 10^−7^	1.27	GCM (θ-solvent)
1000 W-30 min	2.59 ± 0.51 × 10^7^	198 ± 24	137.34	9 ± 10 × 10^−6^	1.44	GCM (θ-solvent)
1200 W-5 min	1.12 ± 0.54 × 10^7^	300 ± 79	142.8	7 ± 4.1 × 10^−5^	2.1	GCP (good-solvent)
1200 W-15 min	1.76 ± 0.27 × 10^7^	178 ± 18	119.49	1.7 ± 1.3 × 10^−5^	1.49	GCM (θ-solvent)
1200 W-30 min	2.01 ± 0.53 × 10^7^	214 ± 33	138.21	1.31 ± 0.85 × 10^−5^	1.55	GCM (θ-solvent)
1400 W-5 min	1.51 ± 0.16 × 10^7^	136 ± 10	115.42	1.5 ± 0.98 × 10^−5^	1.18	Hollow-sphere
1400 W-15 min	2.49 ± 0.26 × 10^7^	159 ± 11	124.75	1.1 ± 0.43 × 10^−5^	1.27	GCM (θ-solvent)
1400 W-30 min	5.7 ± 1.5 × 10^7^	213 ± 32	129.22	2.51 ± 0.26 × 10^−5^	1.65	GCP (θ-solvent)
1600 W-5 min	6.9 ± 1.8 × 10^6^	169 ± 28	128.97	1.3 ± 5 × 10^−5^	1.31	GCM (θ-solvent)
1600 W-15 min	2.3 ± 1 × 10^7^	224 ± 58	137.19	3.8 ± 1.6 × 10^−5^	1.63	GCP (θ-solvent)
1600 W-30 min	2.52 ± 0.44 × 10^7^	200 ± 21	133.82	1.41 ± 0.58 × 10^−5^	1.49	GCM (θ-solvent)
1800 W-5 min	7.8 ± 1.2 × 10^6^	148 ± 16	125.61	8 ± 120 × 10^−7^	1.18	Hollow-sphere
1800 W-15 min	1.62 ± 0.31 × 10^7^	209 ± 24	130.15	3.2 ± 1.4 × 10^−5^	1.61	GCP (θ-solvent)
1800 W-30 min	3.32 ± 0.95 × 10^7^	223 ± 37	129.97	2.29 ± 0.65 × 10^−5^	1.72	GCP (θ-solvent)
2000 W-5 min	3.32 ± 0.57 × 10^6^	120 ± 17	129.56	−5.6 ± 3 × 10^−5^	0.93	Hollow-sphere
2000 W-15 min	7 ± 2.2 × 10^6^	173 ± 36	137.78	−1.1 ± 1.8 × 10^−5^	1.26	GCM (θ-solvent)
2000 W-30 min	2.67 ± 0.74 × 10^7^	227 ± 36	138.9	1.5 ± 0.65 × 10^−5^	1.63	GCP (θ-solvent)

Weight-average molecular weight (M_w_), z-average mean radius of gyration (R_g_), hydrodynamic radius (R_h_), the second virial coefficient (A_2_), Gaussian coil, monodisperse (GCM), Gaussian coil, polydisperse (GCP).

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
