# Peer review of "Effects of High-Intensity Ultrasound Pretreatment on Structure, Properties, and Enzymolysis of Soy Protein Isolate"

_molecules, 2019, doi:10.3390/molecules24203637_

Round 1

Reviewer 1 Report

The research is well addressed, to study the effects of high-intensity ultrasound (HIU) on the structure and properties of soy protein isolate (SPI). However, there are some remarks.

- References 24 and 25 are missing in the introduction, which should be included. Also, the authors should emphasize the novelty of this research.

- SPI hydrolysates had not been considered in the title.

- Lines (202-219): figure 2B and 2C should be included in the text. Also, no mention is made of amide A and amide B in the main-text.

- In figure 3, the content of secondary structure of untreated-SPI should be added, which will allow confirming the sentence “compared with untreated SPI the b-sheet and random coil contents of HIU treated SPI increased, while the a-helix and b-turn content decreased” (lines 231-232)

- The conclusions (lines 455-470) are a list of the results; therefore, they should be re-written, projecting the most important findings towards its application.

Author Response

Dear editor, Thank you very much for giving us an opportunity to revise our manuscript again, we appreciate editor and reviewers very much for their positive and constructive comments on our manuscript entitled “Effects of high-intensity ultrasound pretreatment on structure and properties of soy protein isolate” (ID: molecules-598867). We have studied reviewer’s comments carefully and made revision which marked in yellow in the paper according to the comments. Now we submit our revised manuscript for your kind consideration. I am the corresponding author for this paper. If you have any questions, please let me know. I am looking forward to hearing from you soon. Thank you very much for your kind help. Yours sincerely, Wentao Wang & Haizhou Dong Tel: +86 53 8242850 E-mail address: [email protected] (W. T. Wang), [email protected] (H. Z. Dong) Reviewer 1# 1. References 24 and 25 are missing in the introduction, which should be included. Also, the authors should emphasize the novelty of this research. Re: References 24 and 25 have added in the introduction. The novelty of this research has been emphasized. 2. SPI hydrolysates had not been considered in the title. Re: The title have been reviewed. 3. Lines (202-219): figure 2B and 2C should be included in the text. Also, no mention is made of amide A and amide B in the main-text. Re: Figure 2B and 2C had been added in the text. Amide A and amide B were added in the main-text. 4. In figure 3, the content of secondary structure of untreated-SPI should be added, which will allow confirming the sentence “compared with untreated SPI the β-sheet and random coil contents of HIU treated SPI increased, while the α-helix and β-turn content decreased” (lines 231-232). Re: The content of secondary structure of the control was that of untreated-SPI in figure 3. 5. The conclusions (lines 455-470) are a list of the results; therefore, they should be re-written, projecting the most important findings towards its application. Re: The conclusions (lines 455-470) have been re-written.

Reviewer 2 Report

The authors have demonstrated results that regards the effects of HIU treatment performed at various powers (600-2000 W) and lengths of time (5-30 min) on the structure and property of SPI. the paper is written very well in clear manner. 

Just on suggestion for further studies: The hydroxyl-radical scavenging activity can be evaluated by more specific method than that presented herein. The results of this paper are important for the SPI development and could promote comprehensive utilization of SPI only if further study include more specific evaluation of antioxidants properties. 
